# MACE: Higher Order Equivariant Message Passing Neural Networks for Fast and Accurate Force Fields

**Ilyes Batatia**
Engineering Laboratory,
University of Cambridge
Cambridge, CB2 1PZ UK
Department of Chemistry,
ENS Paris-Saclay, Université Paris-Saclay
91190 Gif-sur-Yvette, France
`ilyes.batatia@ens-paris-saclay.fr`

**Dávid Péter Kovács**
Engineering Laboratory,
University of Cambridge
Cambridge, CB2 1PZ UK

**Gregor N. C. Simm**
Engineering Laboratory,
University of Cambridge
Cambridge, CB2 1PZ UK

**Christoph Ortner**
Department of Mathematics
University of British Columbia
Vancouver, BC, Canada V6T 1Z2

**Gábor Csányi**
Engineering Laboratory,
University of Cambridge
Cambridge, CB2 1PZ UK

## Abstract

Creating fast and accurate force fields is a long-standing challenge in computational chemistry and materials science. Recently, several equivariant message passing neural networks (MPNNs) have been shown to outperform models built using other approaches in terms of accuracy. However, most MPNNs suffer from high computational cost and poor scalability. We propose that these limitations arise because MPNNs only pass two-body messages leading to a direct relationship between the number of layers and the expressivity of the network. In this work, we introduce MACE, a new equivariant MPNN model that uses higher body order messages. In particular, we show that using four-body messages reduces the required number of message passing iterations to just *two*, resulting in a fast and highly parallelizable model, reaching or exceeding state-of-the-art accuracy on the rMD17, 3BPA, and AcAc benchmark tasks. We also demonstrate that using higher order messages leads to an improved steepness of the learning curves.

## 1 Introduction

The earliest approaches for creating force fields (interatomic potentials) using machine learning techniques were using local atom-centered symmetric descriptors and feed-forward neural networks [6], Gaussian Process regression[2] or linear regression [44, 47]. The first attempts to use graph neural networks to model the potential energy of atomistic systems had only limited success. The DTNN [42], SchNet [41], HIP-NN [35], PhysNet [48], or DimeNet [20, 29] approaches could only come close to but not improve upon the atomic descriptor-based methods in terms of computational efficiency

36th Conference on Neural Information Processing Systems (NeurIPS 2022).

and accuracy on public benchmarks. Furthermore, most MPNN interatomic potentials use 2-body invariant messages, making them non-universal approximators [38].

The MACE architecture presented here allows for the efficient computation of equivariant messages with high body order. As a result of the increased body order of the messages, only two message passing iterations are necessary to achieve high accuracy - unlike the typical five or six iterations of MPNNs, making it scalable and parallelizable. Finally, our implementation has remarkable computational efficiency, reaching state-of-the-art results on the 3BPA benchmark after 30 mins of training on NVIDIA A100 GPUs.

We summarise our main contributions as follows:

- We introduce MACE, a novel architecture combining equivariant message passing with efficient many-body messages. The MACE models achieve state-of-the-art performance on challenging benchmark tests. They also display greater generalization capabilities over other approaches on extrapolation benchmarks.

- We demonstrate that many-body messages change the power of the empirical power-law of the learning curves. Furthermore, we show experimentally that the addition of equivariant messages only shifts the learning curves but does not change the power law when higher order messages are used.

- We show that MACE does not only outperform previous approaches in terms of accuracy but also does so while being significantly faster to train and evaluate than the previous most accurate models.

## 2 Background

### 2.1 MPNN Interatomic Potentials

MPNNs [22, 9] are a type of graph neural network (GNN, [40, 4, 27, 51]) that parametrises a mapping from a labeled graph to a target space, either a graph or a vector space. When applied to parameterise properties of atomistic structures (materials or molecules), the graph is embedded in 3-dimensional (3D) Euclidean space, where each node represents an atom, and edges connect nodes if the corresponding atoms are within a given distance of each other. We represent the state of each node $i$ in layer $t$ of the MPNN by a tuple

$$\sigma_i^{(t)} = (\boldsymbol{r}_i, z_i, \boldsymbol{h}_i^{(t)}), \tag{1}$$

where $\boldsymbol{r}_i \in \mathbb{R}^3$ is the position of atom $i$, $z_i$ the chemical element, and $\boldsymbol{h}_i^{(t)}$ are its learnable features. A forward pass of the network consists of multiple *message construction*, *update*, and *readout* steps. During message construction, a message $\boldsymbol{m}_i^{(t)}$ is created for each node by pooling over its neighbors:

$$\boldsymbol{m}_i^{(t)} = \bigoplus_{j \in \mathcal{N}(i)} M_t(\sigma_i^{(t)}, \sigma_j^{(t)}), \tag{2}$$

where $M_t$ is a learnable message function and $\bigoplus_{j \in \mathcal{N}(i)}$ is a learnable, permutation invariant pooling operation over the neighbors of atom $i$ (e.g., a sum). In the update step, the message $\boldsymbol{m}_i^{(t)}$ is transformed into new features

$$\boldsymbol{h}_i^{(t+1)} = U_t(\sigma_i^{(t)}, \boldsymbol{m}_i^{(t)}), \tag{3}$$

where $U_t$ is a learnable update function. After $T$ message construction and update steps, the learnable readout functions $\mathcal{R}_t$ map the node states $\sigma_i^{(t)}$ to the target, in this case the site energy of atom $i$,

$$E_i = \sum_{t=1}^{T} \mathcal{R}_t(\sigma_i^{(t)}). \tag{4}$$

### 2.2 Equivariant Graph Neural Networks

In *equivariant* GNNs, internal features $\boldsymbol{h}_i^{(t)}$ transform in a specified way under some group action [1, 12, 32, 46, 49]. When modelling the potential energy of an atomic structure, the group of interest is

O(3), specifying rotations and reflections of the particles.[1] We call a GNN O(3) equivariant if it has internal features that transform under the rotation $Q \in O(3)$ as

$$\boldsymbol{h}_i^{(t)}(Q \cdot (\boldsymbol{r}_1, ..., \boldsymbol{r}_N)) = D(Q)\boldsymbol{h}_i^{(t)}(\boldsymbol{r}_1, ..., \boldsymbol{r}_N), \tag{5}$$

where $Q \cdot (\boldsymbol{r}_1, ..., \boldsymbol{r}_N)$ denotes the action of the rotation on the set of atomic positions and $D(Q)$ is a matrix representing the rotation $Q$, acting on message $\boldsymbol{h}_i^{(t)}$. In general, elements of the feature vector can be labeled according to the irreducible representation they transform with. We will write $h_{i,kLM}^{(t)}$ to indicate a collection of features on atom $i$, indexed by $k$, that transform according to

$$h_{i,kLM}^{(t)}(Q \cdot (\boldsymbol{r}_1, \ldots, \boldsymbol{r}_N)) = \sum_{M'} D_{M'M}^L(Q)h_{i,kLM'}^{(t)}(\boldsymbol{r}_1, \ldots, \boldsymbol{r}_N), \tag{6}$$

where $D^L(Q) \in \mathbb{R}^{(2L+1)\times(2L+1)}$ is a Wigner D-matrix of order $L$. A feature labelled with $L = 0$ describes an invariant scalar. Features labeled with $L > 0$, describe equivariant features, formally corresponding to equivariant vectors, matrices or higher order tensors. The features of *invariant* models, such as SchNet[41] and DimeNet[29], transform according to $D(Q) = \mathbb{1}$, the identity matrix. Models such as NequIP [5], equivariant transformer [45], PaiNN [43], or SEGNNs [8], in addition to invariant scalars, employ equivariant internal features that transform like vectors or tensors.

## 3 Related Work

**ACE - Higher Order Local Descriptors**    In the last few years, there have been two significant breakthroughs in machine learning force fields. First, the Atomic Cluster Expansion (ACE) [16] provided a systematic framework for constructing high body order complete polynomial basis functions (features) at a constant cost per basis function, independent of body order [17]. It has also been shown that ACE includes many previously developed atomic environment representations as special cases, including Atom Centred Symmetry Functions [6], the Smooth Overlap of Atomic Positions (SOAP) descriptor [2], Moment Tensor Potential basis functions [44], and the hyperspherical bispectrum descriptor [2] used by the SNAP model [47]. These local models are limited by their cutoff distance and their relatively rigid architecture compared to the overparametrised MPNNs, leading to somewhat lower accuracy, in particular, for molecular force fields.

**Equivariant MPNNs**    The second breakthrough was using equivariant internal features in MPNNs. These equivariant MPNNs, such as Cormorant [1], Tensor Field Networks [46], EGNN [39], PaiNN [43], Equivariant Transformers [45], SEGNN [8], NewtonNet [23], and NequIP [5] were able to achieve higher performance than previous local descriptor-based models. However, they suffer from two significant limitations: first, the most accurate models used $L = 3$ spherical tensors as messages and 4 to 6 message passing iterations [5], which resulted in a relatively high computational cost. Second, using this many iterations significantly increased the receptive field of the network, making them difficult to parallelise across multiple GPUs [36].

**Higher Order Message Passing**    Most MPNNs use a message passing scheme based on two-body messages, meaning they simultaneously depend on the states of two atoms. It has been recognised that it can be beneficial to include angles into the features, effectively creating 3-body invariant messages [29]. This idea has also been exploited in other invariant MPNNs, in particular, by SphereNet [34] and GemNet [30]. Even though these models improved the accuracy compared to the 2-body message passing, they were limited by the computational cost associated with explicitly summing over all triplets or quadruplets to compute the higher order features.

**Multi-ACE Framework**    Recently, multi-ACE has been proposed as a unifying framework of $E(3)$-equivariant atom-centered interatomic potentials, extending the ACE framework to include methods built on equivariant MPNNs [3]. A similar unifying theories were also put forward by [37] and [7]. The idea is to parameterise the message $\boldsymbol{m}_i^{(t)}$ in terms of invariant or equivariant ACE models. This framework sets out a design space in which each model can be characterised in terms of: (1) the number of layers, (2) the body order of the messages, (3) the equivariance (or invariance) of the messages, and (4) the number of features in each layer. The framework highlights the relationship between the overall body order of the models and message passing, also previously

---

[1]Translation invariance is trivially incorporated through the use of relative distances.

discussed in Kondor [31]. Most previously published models achieved high accuracy by *either* using 4 to 6 layers [5, 43] *or* increasing the local body order with a single layer [33, 36]. With our model, we fall in between these two extremes by combining high body order with message passing.

## 4 The MACE Architecture

Our MACE model follows the general framework of MPNNs outlined in Section 2. Our key innovation is a new message construction mechanism. We expand the messages $\boldsymbol{m}_i^{(t)}$ in a hierarchical body order expansion,

$$\boldsymbol{m}_i^{(t)} = \sum_j \boldsymbol{u}_1\left(\sigma_i^{(t)}; \sigma_j^{(t)}\right) + \sum_{j_1, j_2} \boldsymbol{u}_2\left(\sigma_i^{(t)}; \sigma_{j_1}^{(t)}, \sigma_{j_2}^{(t)}\right) + \cdots + \sum_{j_1, \ldots, j_\nu} \boldsymbol{u}_\nu\left(\sigma_i^{(t)}; \sigma_{j_1}^{(t)}, \ldots, \sigma_{j_\nu}^{(t)}\right),$$
(7)

where the $\boldsymbol{u}$ functions are learnable, the sums run over the neighbors of $i$, and $\nu$ is a hyper-parameter corresponding to the maximum correlation order, the body order minus 1, of the message function with respect to the states. Even though we refer to the message as $(\nu + 1)$-body with respect to the states, the overall body order with respect to the positions can be larger depending on the body order of the states themselves. Crucially, by writing $\sum_{j_1, \ldots, j_\nu}$, which includes self-interaction (e.g., $j_1 = j_2$), we will later obtain a tensor product structure with a computationally efficient parameterisation, that allows us to circumvent the seemingly exponential scaling of the computational cost with the correlation order $\nu$. This contrasts with previous models, such as DimeNet [28, 29], that compute 3-body features via the more standard many-body expansion $\sum_{j_1 < \cdots < j_\nu}$. Below, we describe the MACE architecture in detail. To better understand the architecture, we report in A.4 a table of the introduced tensors along with their shapes.

**Message Construction** At each iteration, we first embed the edges using a learnable radial basis $R_{kl_1l_2l_3}^{(t)}$, a set of spherical harmonics $Y_{l_1}^{m_1}$, and a learnable embedding of the previous node features $h_{j,\tilde{k}l_2m_2}^{(t)}$ using weights $W_{k\tilde{k}l_2}^{(t)}$. The $\boldsymbol{A}_i^{(t)}$-features are obtained by pooling over the neighbours $\mathcal{N}(i)$ to obtain permutation invariant 2-body features whilst, crucially, retaining full directional information, and thus, full information about the atomic environment:

$$A_{i,kl_3m_3}^{(t)} = \sum_{l_1m_1,l_2m_2} C_{l_1m_1,l_2m_2}^{l_3m_3} \sum_{j \in \mathcal{N}(i)} R_{kl_1l_2l_3}^{(t)}(r_{ji}) Y_{l_1}^{m_1}(\hat{\boldsymbol{r}}_{ji}) \sum_{\tilde{k}} W_{k\tilde{k}l_2}^{(t)} h_{j,\tilde{k}l_2m_2}^{(t)},$$
(8)

where $C_{l_1m_1,l_2m_2}^{l_3m_3}$ are the standard Clebsch-Gordan coefficients ensuring that $A_{i,kl_3m_3}^{(t)}$ maintain the correct equivariance, $r_{ji}$ is the (scalar) interatomic distance, and $\hat{\boldsymbol{r}}_{ji}$ is the corresponding unit vector. $R_{kl_1l_2l_3}^{(t)}$ is obtained by feeding a set of radial features that embed the radial distance $r_{ji}$ using Bessel functions multiplied by a smooth polynomial cutoff (cf. Ref. [29]) to a multi-layer perceptron (MLP). See Section A.5 for details. In the first layer, the node features $h_j^{(t)}$ correspond to the (invariant) chemical element $z_j$. Therefore, (8) can be further simplified:

$$A_{i,kl_1m_1}^{(1)} = \sum_{j \in \mathcal{N}(i)} R_{kl_1}^{(1)}(r_{ji}) Y_{l_1}^{m_1}(\hat{\boldsymbol{r}}_{ji}) W_{kz_j}^{(1)}.$$
(9)

This simplified operation is much cheaper, making the computational cost of the first layer low.

The *key* operation of MACE is the efficient construction of higher order features from the $\boldsymbol{A}_i^{(t)}$-features. This is achieved by first forming tensor products of the features, and then symmetrising:

$$\boldsymbol{B}_{i,\eta_\nu kLM}^{(t)} = \sum_{\boldsymbol{lm}} \mathcal{C}_{\eta_\nu,\boldsymbol{lm}}^{LM} \prod_{\xi=1}^{\nu} \sum_{\tilde{k}} w_{k\tilde{k}l_\xi}^{(t)} A_{i,\tilde{k}l_\xi m_\xi}^{(t)}, \quad \boldsymbol{lm} = (l_1m_1, \ldots, l_\nu m_\nu)$$
(10)

where the coupling coefficients $\mathcal{C}_{\eta_\nu}^{LM}$ corresponding to the generalised Clebsch-Gordan coefficients (details in A.3) ensuring that $\boldsymbol{B}_{i,\eta_\nu kLM}^{(t)}$ are $L$-equivariant, the weights $w_{k\tilde{k}l_\xi}^{(t)}$ are mixing the channels $(k)$ of $\boldsymbol{A}_i^{(t)}$, and $\nu$ is a given correlation order. $\mathcal{C}_{\eta_\nu,\boldsymbol{lm}}^{LM}$ is very sparse and can be pre-computed such that (10) can be evaluated efficiently (see Appendix A.3.3). The additional index $\eta_\nu$ simply enumerates all possible couplings of $l_1, \ldots, l_\nu$ features that yield the selected equivariance specified

by the $L$ index. The $\boldsymbol{B}_i^{(t)}$-features are constructed up to some maximum $\nu$. This variable in (10) is the order of the tensor product, and hence, can be identified as the order of the many-body expansion terms in (7). The computationally expensive multi-dimensional sums over all triplets, quadruplets, etc., are thus circumvented and absorbed into (9) and (8).

The message $\boldsymbol{m}_i^{(t)}$ can now be written as a linear expansion

$$m_{i,kLM}^{(t)} = \sum_{\nu} \sum_{\eta_\nu} W_{z_i kL, \eta_\nu}^{(t)} \boldsymbol{B}_{i,\eta_\nu kLM}^{(t)}, \tag{11}$$

where $W_{z_i kL, \eta_\nu}^{(t)}$ is a learnable weight matrix that depends on the chemical element $z_i$ of the receiving atom and message symmetry $L$. Thus, we implicitly construct each term $\boldsymbol{u}$ in (7) by a linear combination of $\boldsymbol{B}_{i,\eta_\nu kLM}^{(t)}$ features of the corresponding body order.

Under mild conditions on the two-body bases $\boldsymbol{A}_i^{(t)}$, the higher order features $\boldsymbol{B}_{i,\eta_\nu kLM}^{(t)}$ can be interpreted as a *complete basis* of many-body interactions [17], which can be computed at a cost comparable to pairwise interactions. Because of this, the expansion (11) is *systematic*. It can in principle be converged to represent any smooth $(\nu + 1)$-body equivariant mapping in the limit of infinitely many features (proof in [17]).

**Update**    In MACE, the update is a linear function of the message and the residual connection [25]:

$$h_{i,kLM}^{(t+1)} = U_t^{kL}(\sigma_i^{(t)}, \boldsymbol{m}_i^{(t)}) = \sum_{\tilde{k}} W_{kL,\tilde{k}}^{(t)} m_{i,\tilde{k}LM}^{(t)} + \sum_{\tilde{k}} W_{z_i kL,\tilde{k}}^{(t)} h_{i,\tilde{k}LM}^{(t)}. \tag{12}$$

**Readout**    In the readout phase, the invariant part of the node features is mapped to a hierarchical decomposition of site energies via readout functions:

$$E_i = E_i^{(0)} + E_i^{(1)} + ... + E_i^{(T)}, \qquad \text{where}$$

$$E_i^{(t)} = \mathcal{R}_t\left(\boldsymbol{h}_i^{(t)}\right) = \begin{cases} \sum_{\tilde{k}} W_{\text{readout},\tilde{k}}^{(t)} h_{i,\tilde{k}00}^{(t)} & \text{if } t < T \\ \text{MLP}_{\text{readout}}^{(t)}\left(\left\{h_{i,k00}^{(t)}\right\}_k\right) & \text{if } t = T \end{cases} \tag{13}$$

The readout only depends on the invariant features $h_{i,k00}^{(t)}$ to ensure that the site energy contributions $E_i^{(t)}$ are invariant as well. To maintain body ordering, we use linear readout functions for all layers except the last, where we use a one-layer MLP.

## 5    Results

### 5.1    Effect of Higher Order Messages

**Number of Layers**    In this section, we investigate the effect of using higher order messages. Many MPNN architectures [41, 48] exclusively pass two-body invariant messages resulting in an incomplete representation of the local environment [38]. Equivariant message-passing schemes [5, 43, 8] lift the degeneracy of most structures by containing directional information in the messages. MPNNs that only employ two-body messages at each layer can increase the body order *either* by stacking layers [31] which simultaneously increases the model's receptive field *or* by using non-linear activation functions, generate only a subset of all possible higher order features. By constructing higher order messages using the MACE architecture, we disentangle the increase in body order from the increase of the receptive field.

In Figure 1, we show the accuracy of MACE, NequIP, and BOTNet [3] on the 3BPA benchmark [33] as a function of the number of message passing layers. Approaches employing 2-body message passing require up to five iterations for their accuracy to converge. By constructing many body messages, the number of required layers to converge in accuracy reduces to just two. In all subsequent experiments, we use two-layer MACE models.

Furthermore, we compare BOTNet, which does not use any non-linearities in the update step to NequIP, which does. Otherwise, the two models are very similar. We observe that the increase in body

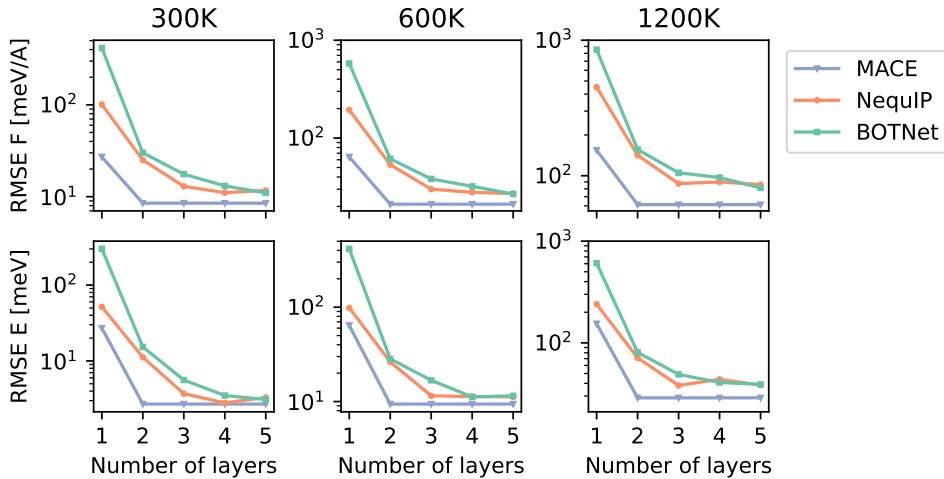

Figure 1: Energy and force errors of BOTNet, NequIP, and MACE ($L = 2$) on the 3BPA dataset at different temperatures as a function of the number of layers.

order through non-linearities within the update provides only marginal improvement, highlighting the difference between an increase in body order through non-linearities (NequIP) and higher order symmetric messages (MACE). Consequently, higher order message passing allows one to reduce the number of layers, thereby increasing speed and ease of parallelization over multiple GPUs. We note that MACE does not improve after two layers as the diameter of the 3BPA molecule is about 9 Å and radial cutoff in each layer is 5 Å.

**Learning Curves** We study how higher order message passing affects the learning curves. A recent study of the NequIP model [5] showed that the inclusion of equivariant features results in enhanced data efficiency, increasing the slope of the log-log plot of predictive error as a function of the dataset size. They showed that adding equivariance not only *shifts* the learning curves, but also changes the powers in the empirical power law of the learning curves, which is usually constant for a given dataset [26].

On the left panel of Figure 2, we replicate the experiments of [5] by training a series of *invariant* MACE models with increasing correlation order $\nu$ on the aspirin molecule from the rMD17 dataset. We observe that adding higher order messages changes the steepness of the learning curves, even without equivariant messages. The model with correlation order $\nu = 1$ corresponds to a two-layer 2-body invariant model, similar to SchNet. This model is the least accurate due to the incomplete nature of 2-body invariant representations of the local environment [38]. The invariant messages with $\nu = 2$ are akin to those in DimeNet, which explicitly puts angular information into the messages. We see that including higher order information significantly improves the model's accuracy. Finally, by going beyond any current message passing potential by setting $\nu = 3$, we achieve similar performance to a highly-accurate 2-body, equivariant MPNN while only using higher order invariant messages.

On the middle panel of Figure 2, we keep the correlation order fixed at $\nu = 3$ and gradually increase the symmetry order $L$ of the messages. While the slope remains nearly unchanged, the curves are shifted. In the right panel of Figure 2, we keep the correlation order fixed at $\nu = 1$ and gradually increase the symmetry order $L$ of the messages. We see only a marginal slope change when adding equivariant features, which could be attributable to the relatively low expressiveness of a two-layer MACE restricted to correlation order $\nu = 1$. These results suggest two routes to improve invariant 2-body MPNN models: creating higher correlation order messages or incorporating equivariant messages. By exploiting both of these options, the MACE model achieves state-of-the-art accuracy.

## 5.2 Scaling and Computational Cost

**Chemical Elements** A significant limitation of existing atomic environment representations is that their size grows with the number of chemical elements $S$ and correlation order $\nu$ as $S^\nu$. Data-driven compression schemes have been proposed [50] to solve this issue, and MPNNs incorporate similar

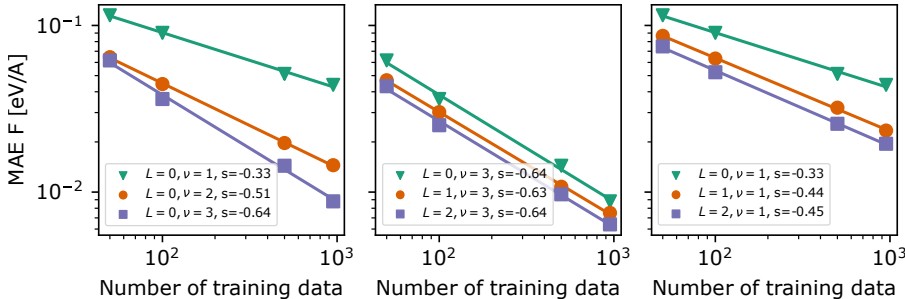

Figure 2: Learning curve of force errors (MAE in eV / Å) for aspirin from the rMD17 dataset for different models. *Left:* Two layers of invariant ($L = 0$) MACE with increasing body order $\nu \in \{1, 2, 3\}$. *Center:* Two layers of MACE with $\nu = 3$ and increasing equivariance $L \in \{0, 1, 2\}$. *Right:* Two layers of MACE with $\nu = 1$ and increasing equivariance $L \in \{0, 1, 2\}$. In each case the slope ($s$) is indicated.

embeddings of the chemical elements into a fixed-size vector space. MACE uses a continuous species embedding and when constructing the higher order features in (10), it does not include the species dimension $k$ in the tensor product resulting in $\mathcal{O}(1)$ scaling of the model with the number of chemical elements $S$.

**Receptive Field**     A severe limitation of many previously published MPNNs was their large receptive field, making it difficult to parallelize the evaluation across multiple GPUs. In traditional MPNNs, the total receptive field of each node, which grows with each message passing iteration, can be up to 30 Å. This scaling results in the number of neighbours being in the thousands in a condensed phase simulation, preventing any efficient parallelization [36]. By decoupling the increase in correlation order of the messages from the number of message passing iterations, MACE only requires two layers resulting in a much smaller receptive field. With a local radial cutoff of 4 to 5 Å, the overall receptive field remains small, making the model more parallelisable.

**Computational Cost**     The computational bottleneck of equivariant MPNNs is the equivariant tensor product (8). This tensor product is evaluated on edges. In MACE, we only evaluate this expensive tensor product once, within the second layer, and build up correlations through the tensor product of (10). Importantly, this operation is carried out on nodes. Typically the number of nodes is orders of magnitudes smaller than the number of edges resulting in a computational advantage. In addition, we developed a loop tensor contraction algorithm for the efficient implementation of (10) and (11) detailed in Section A.3.

We report evaluation times for BOTNet, NequIP, and multiple versions of MACE in Table 2. We observe that the invariant MACE ($L = 0$) is close to 10 times faster than BOTNet and NequIP while achieving similar accuracy at high temperatures. MACE with $L = 1$ and $L = 2$ is 5 and 4 times faster than BOTNet and NequIP, respectively, while outperforming them at every temperature. We acknowledge that accurate speed comparisons between codes are hard to obtain, and further investigations need to be carried out. It is also essential to consider training times. In order to do a fair comparison, all the timings were realised using the `mace` code that implements all the above models. Models that are significantly faster to train are better suited for applications of active learning, which is typically how databases for materials science applications are built [13–15]. The MACE model reported in Table 2 takes approximately 30 mins to reach the accuracy of a converged BOTNet model, taking more than a day to be trained on the 3BPA dataset using NVIDIA A100 GPUs.

## 5.3    Benchmark Results [2]

### 5.3.1    rMD17: Molecular Dynamics Trajectory

The revised MD17 (rMD17) dataset contains train test splits randomly selected from a long molecular dynamics trajectory of ten small organic molecules [11]. For each molecule, the splits consist of 1000 training and test configurations. Table 1 shows that MACE achieves excellent accuracy, improving

---

[2]Training details and hyper-parameters for all experiments can be found in Appendix A.5

Table 1: **Mean absolute errors on the rMD17 dataset** [11]. Energy (E, meV) and force (F, meV/Å) errors of different models trained on 950 configurations and validated on 50. The models on the right of the first vertical line, DimeNet and NewtonNet, were trained on the original MD17 dataset [10]. The models on the right of the second (double) vertical line were trained on just 50 configurations.

| | | MACE | Allegro [36] | BOTNet [3] | NequIP [5] | GemNet (T/Q) [30] | ACE [33] | FCHL [18] | GAP [2] | ANI [19] | PaiNN [43] | DimeNet [29] | NewtonNet [24] | ACE [33] | NequIP [5] | MACE |
|---|---|---|---|---|---|---|---|---|---|---|---|---|---|---|---|---|
| | | | | | | | $N_{\text{train}} = 1000$ | | | | | | | | $N_{\text{train}} = 50$ | |
| Aspirin | E | **2.2** | 2.3 | 2.3 | 2.3 | - | 6.1 | 6.2 | 17.7 | 16.6 | 6.9 | 8.8 | 7.3 | 26.2 | 19.5 | **17.0** |
| | F | **6.6** | 7.3 | 8.5 | 8.2 | 9.5 | 17.9 | 20.9 | 44.9 | 40.6 | 16.1 | 21.6 | 15.1 | 63.8 | 52.0 | 43.9 |
| Azobenzene | E | 1.2 | 1.2 | **0.7** | **0.7** | - | 3.6 | 2.8 | 8.5 | 15.9 | - | - | 6.1 | 9.0 | 6.0 | 5.4 |
| | F | 3.0 | **2.6** | 3.3 | 2.9 | - | 10.9 | 10.8 | 24.5 | 35.4 | - | - | 5.9 | 28.8 | 20.0 | 17.7 |
| Benzene | E | 0.4 | 0.3 | **0.03** | 0.04 | - | 0.04 | 0.35 | 0.75 | 3.3 | - | 3.4 | - | **0.2** | 0.6 | 0.7 |
| | F | 0.3 | **0.2** | 0.3 | 0.3 | 0.5 | 0.5 | 2.6 | 6.0 | 10.0 | - | 8.1 | - | **2.7** | 2.9 | **2.7** |
| Ethanol | E | **0.4** | **0.4** | **0.4** | **0.4** | - | 1.2 | 0.9 | 3.5 | 2.5 | 2.7 | 2.8 | 2.6 | 8.6 | 8.7 | 6.7 |
| | F | **2.1** | **2.1** | 3.2 | 2.8 | 3.6 | 7.3 | 6.2 | 18.1 | 13.4 | 10.0 | 10.0 | 9.1 | 43.0 | 40.2 | 32.6 |
| Malonaldehyde | E | 0.8 | **0.6** | 0.8 | 0.8 | - | 1.7 | 1.5 | 4.8 | 4.6 | 3.9 | 4.5 | 4.1 | 12.8 | 12.7 | **10.0** |
| | F | 4.1 | **3.6** | 5.8 | 5.1 | 6.6 | 11.1 | 10.3 | 26.4 | 24.5 | 13.8 | 16.6 | 14.0 | 63.5 | 52.5 | 43.3 |
| Naphthalene | E | 0.5 | **0.2** | **0.2** | 0.9 | - | 0.9 | 1.2 | 3.8 | 11.3 | 5.1 | 5.3 | 5.2 | 3.8 | **2.1** | **2.1** |
| | F | 1.6 | 0.9 | 0.9 | 1.3 | 1.9 | 5.1 | 6.5 | 16.5 | 29.2 | 3.6 | 9.3 | 3.6 | 19.7 | 10.0 | 9.2 |
| Paracetamol | E | **1.3** | 1.5 | **1.3** | 1.4 | - | 4.0 | 2.9 | 8.5 | 11.5 | - | - | 6.1 | 13.6 | 14.3 | 9.7 |
| | F | **4.8** | 4.9 | 5.8 | 5.9 | - | 12.7 | 12.3 | 28.9 | 30.4 | - | - | 11.4 | 45.7 | 39.7 | 31.5 |
| Salicylic acid | E | 0.9 | 0.9 | 0.8 | **0.7** | - | 1.8 | 1.8 | 5.6 | 9.2 | 4.9 | 5.8 | 4.9 | 8.9 | 8.0 | 6.5 |
| | F | 3.1 | **2.9** | 4.3 | 4.0 | 5.3 | 9.3 | 9.5 | 24.7 | 29.7 | 9.1 | 16.2 | 8.5 | 41.7 | 35.0 | 28.4 |
| Toluene | E | 0.5 | 0.4 | **0.3** | **0.3** | - | 1.1 | 1.7 | 4.0 | 7.7 | 4.2 | 4.4 | 4.1 | 5.3 | 3.3 | **3.1** |
| | F | **1.5** | 1.8 | 1.9 | 1.6 | 2.2 | 6.5 | 8.8 | 17.8 | 24.3 | 4.4 | 9.4 | 3.8 | 27.1 | 15.1 | 12.1 |
| Uracil | E | 0.5 | 0.6 | **0.4** | **0.4** | - | 1.1 | 0.6 | 3.0 | 5.1 | 4.5 | 5.0 | 4.6 | 6.5 | 7.3 | 4.4 |
| | F | 2.1 | **1.8** | 3.2 | 3.1 | 3.8 | 6.6 | 4.2 | 17.6 | 21.4 | 6.1 | 13.1 | 6.4 | 36.2 | 40.1 | 25.9 |

the state of the art for some molecules, particularly those with the highest errors. As several methods achieve similar accuracy on the standard task of predicting energies and forces based on the whole training set, we also trained MACE and NequIP, another accurate model, on just 50 configurations to increase the difficulty of the benchmark. In this case, we found that MACE outperformed NequIP for most molecules.

### 5.3.2 3BPA: Extrapolation to Out-of-domain Data

The 3BPA dataset introduced in [33] tests a model's extrapolation capabilities. Its training set contains 500 geometries sampled from 300 K molecular dynamics simulation of the large and flexible drug-like molecule 3-(benzyloxy)pyridin-2-amine. The three test sets contain geometries sampled at 300 K, 600 K, and 1200 K to assess in- and out-of-domain accuracy. A fourth test set consists of optimized geometries, where two of the molecule's dihedral angles are fixed, and a third is varied between 0 and 360 degrees resulting in so-called *dihedral slices* through regions of the PES far away from the training data.

The root-mean-squared errors (RMSE) on energies and forces for several models are shown in Table 2. It can be seen that MACE outperforms the other models on all tasks. In particular, when extrapolating to 1200 K data, MACE with $L = 2$ outperforms NequIP and Allegro models by about $30\%$. Further, MACE with $L = 2$ outperforms the next best model, BOTNet, by $40\%$ on energies for the dihedral slices. Finally, the MACE model with invariant messages ($L = 0$) often nearly matches or exceeds the performance of competitive equivariant models.

Table 2: **Root-mean-square errors on the 3BPA dataset.** Energy (E, meV) and force (F, meV/Å) errors of models trained and tested on configurations collected at 300 K of the flexible drug-like molecule 3-(benzyloxy)pyridin-2-amine (3BPA). Standard deviations are computed over three runs and shown in brackets if available. In order to facilitate measuring the efficiency of *architectures* we implemented the NequIP and BOTNet architectures in the same code that we used for MACE and which is published together with this paper. For the precise specification of our NequIP implementation see the Appendix A.5.2. All PyTorch timings were realised on an NVIDIA A100 GPU custom implementations.

| | | Allegro (L=3) | NequIP (L=3) | NequIP (L=3) | BOTNet (L=3) | MACE (L=0) | MACE (L=1) | MACE (L=2) |
|---|---|---|---|---|---|---|---|---|
| Code | | allegro [36] | nequip [5] | mace | mace | mace | mace | mace |
| 300 K | E | 3.84 (0.08) | 3.3 (0.1) | 3.1 (0.1) | 3.1 (0.13) | 4.5 (0.25) | 3.4 (0.2) | **3.0** (0.2) |
| | F | 12.98 (0.17) | 10.8 (0.2) | 11.3 (0.2) | 11.0 (0.14) | 14.6 (0.5) | 10.3 (0.3) | **8.8** (0.3) |
| 600 K | E | 12.07 (0.45) | 11.2 (0.1) | 11.3 (0.31) | 11.5 (0.6) | 13.7 (0.16) | 9.9 (0.8) | **9.7** (0.5) |
| | F | 29.17 (0.22) | 26.4 (0.1) | 27.3 (0.3) | 26.7 (0.29) | 33.3 (1.35) | 24.6 (1.1) | **21.8** (0.6) |
| 1200 K | E | 42.57 (1.46) | 38.5 (1.6) | 40.8 (1.3) | 39.1 (1.1) | 37.1 (0.8) | 31.7 (0.5) | **29.8** (1.0) |
| | F | 82.96 (1.77) | 76.2 (1.1) | 86.4 (1.5) | 81.1 (1.5) | 81.6 (3.89) | 67.8 (1.8) | **62.0** (0.7) |
| Dihedral Slices | E | - | - | 23.2 | 16.3 (1.5) | 12.3 (0.8) | 11.5 (0.6) | **7.8** (0.6) |
| | F | - | - | 23.1 | 20.0 (1.2) | 26.1 (2.8) | 19.3 (0.6) | **16.5** (1.7) |
| Time latency [ms] | | - | - | 103.5 | 101.2 | **10.5** | 17.5 | 24.3 |

MACE shows excellent results while also featuring low computational cost compared to many other models. The $L = 0$ model, which approaches previous models in terms of accuracy, outpaces them by nearly a factor of 10, whereas the $L = 2$ model achieves state-of-the-art accuracy and is around four times faster than other equivariant MPNN models. In the table, we characterise the evaluation speed of the models by reporting the "latency" which is defined as the time it takes to compute forces on a structure, which is typically independent of the number of atoms until GPU threads are filled (typically 10,000 atoms for these models on an Nvidia A100 80GB GPU).

In Figure 3, we compare the BOTNet, NequIP, and MACE ($L = 2$) by inspecting their energy profile for three dihedral slices. Overall, it can be seen that all models produce smooth energy profiles and that, in general, MACE comes closest to the ground truth. The fact that MACE outperforms the

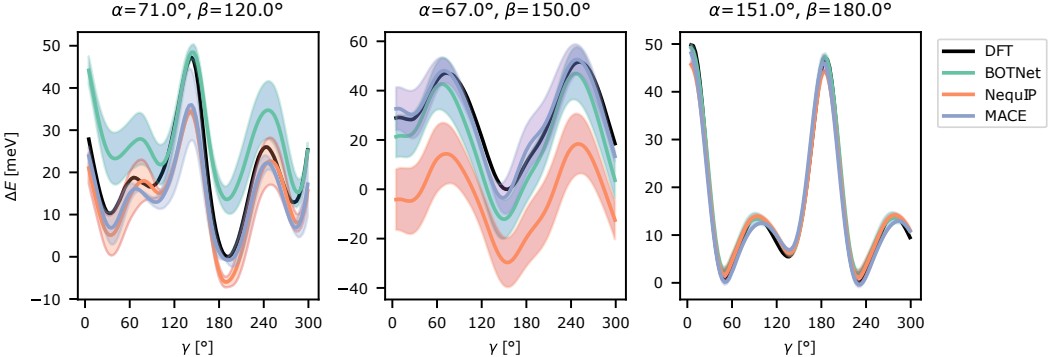

Figure 3: Energy predictions on three cuts through the potential energy surface of the 3-(benzyloxy)pyridin-2-amine (3BPA) molecule by BOTNet, NequIP, and MACE ($L = 2$). The ground-truth energy (DFT) is shown in black. For each cut, the curves have been shifted vertically so that the lowest ground-truth energy is zero.

other methods in the middle panel, which contains geometries furthest from the training dataset [3], suggests superior extrapolation capabilities.

### 5.3.3   AcAc: Flexibility and Reactivity

A similar benchmark dataset assessing a model's extrapolation capabilities to higher temperatures, bond breaking, and bond torsions of the acetylacetone molecule was proposed in [3]. In Table 3, we show that MACE achieves state-of-the-art results on this dataset as well. For details, see Appendix 5.3.3.

Table 3: **Root-mean-square errors on the acetylacetone dataset.** Energy (E, meV) and force (F, meV/Å) errors of models trained on configurations of the acetylacetone molecule sampled at 300 K and tested on configurations sampled at 300 K and 600 K. Standard deviations are computed over three runs.

|  |  | **BOTNet** | **NequIP** | **MACE** |
|---|---|---|---|---|
| 300 K | E | 0.89 (0.0) | **0.81 (0.04)** | 0.9 (0.03) |
|  | F | 6.3 (0.0) | 5.90 (0.38) | **5.1** (0.10) |
| 600 K | E | 6.2 (1.1) | 6.04 (1.26) | **4.6** (0.3) |
|  | F | 29.8 (1.0) | 27.8 (3.29) | **22.4** (0.9) |
| N° Parameters |  | 2,756,416 | 3,190,488 | 2,803,984 |

## 6   Discussions

With MACE, we extend traditional (equivariant) MPNNs from 2-body to many-body message passing in a computationally efficient manner. Our experiments show that the approach reduces the required

number of message passing, leading to efficient and parallelizable models. Furthermore, we have demonstrated the high accuracy and good extrapolation capabilities of MACE, reaching state-of-the-art accuracy on the rMD17, 3BPA, and AcAc benchmarks. Future development should concentrate on testing MACE on larger systems, including condensed phases and solids.

## 7 Reproducibility Statements

We have included error bars via different seeds and various ablation studies wherever necessary and appropriate. We have stated all hyper-parameters and data description in the Appendix A.5. Source code is available at `https://github.com/ACEsuit/mace`.

## 8 Ethical Statements

The societal impact of MACE is challenging to predict. However, better force fields have a positive impact on society by speeding up drug discovery and through helping to understand, control, and design new materials. However, machine learning force fields rely on generating *ab initio* training data leading to heavy computation and large energy consumption. Machine learned force fields do alleviate the costs of doing molecular modelling significantly when compared with using *solely ab initio* methods.

## Acknowledgments and Disclosure of Funding

## 9 Acknowledgement

The authors acknowledge useful discussions with Simon Batzner, Albert Musaelian and William Baldwin. This work was performed using resources provided by the Cambridge Service for Data Driven Discovery (CSD3) operated by the University of Cambridge Research Computing Service (www.csd3.cam.ac.uk), provided by Dell EMC and Intel using Tier-2 funding from the Engineering and Physical Sciences Research Council (capital grant EP/T022159/1), and DiRAC funding from the Science and Technology Facilities Council (www.dirac.ac.uk). DPK acknowledges support from AstraZeneca and the Engineering and Physical Sciences Research Council. CO is supported by Leverhulme Research Project Grant RPG-2017-191 and by the Natural Sciences and Engineering Research Council of Canada (NSERC) [funding reference number IDGR019381].

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
