# A   Appendix

## A.1   Acetylacetone Dataset: Additional Experiments

We ran additional experiments with the acetylacetone dataset introduced in [3] to further investigate the generalization capabilities of MACE [3]. Figure 4 shows the energy predictions of BOTNet [3], NequIP [5], MACE, and (linear) ACE [33] for two trajectories on the acetylacetone's potential energy surface (PES). The left panel shows the energy profile for a rotation around an O-C-C-C dihedral angle. Since the training set only contains dihedral angles below 30° (see lower panel), accurate predictions for angles up to 180° require significant extrapolation capabilities. Also the energy barrier of the rotation is with 1 eV well outside the energy range of the training set which is sampled at 300 K. It can be seen that all models solve this task surprisingly well.

In the right panel of Figure 4, we show energy predictions along a minimum energy path of an intramolecular hydrogen transfer reaction. This task probes a model's ability to describe a bond breaking reaction, something it has not seen in the training data. It should be noted that this reaction occurs in a region of the PES that is not too far from the training data as can be seen from the histogram below. All models accurately reproduce the barrier's shape with the MPNN models closely matching the barrier height as well.

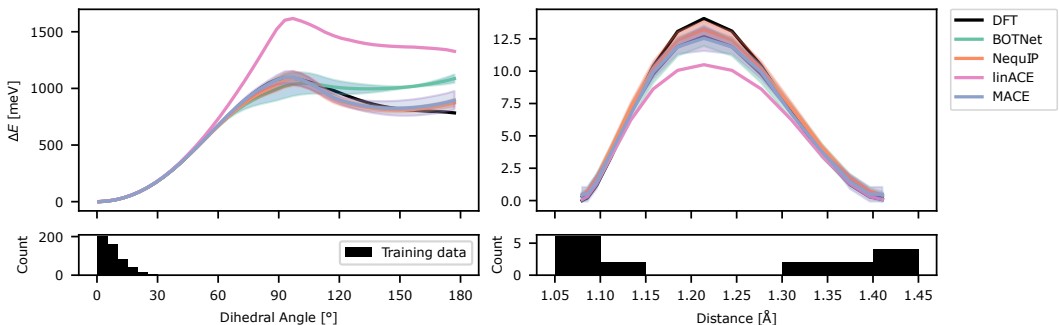

Figure 4:   *Left:* energy predictions for a dihedral slice of the DFT potential energy surface of acetylacetone. *Right:* energy predictions for the proton transfer in acetylacetone. Error bars indicate one standard deviation computed over three runs. The histograms show the distribution of the training data along the relevant coordinate.

## A.2   Description of the Datasets

### A.2.1   rMD17 Dataset

The revised MD17 (rMD17) dataset contains five train test splits of 10 different small organic molecules [11]. Each of the splits contains 1000 configurations for each molecule sampled randomly from a long *Ab initio* molecular dynamics simulation carried out at 500 K computed with DFT. We note that the older version of the dataset, called MD17, has been shown to contain noisy labels [11].

### A.2.2   3BPA Dataset

The 3BPA dataset contains DFT train test splits of a flexible drug-like organic molecule sampled from different temperature molecular dynamics trajectories [33]. The models is trained on 500 snapshots sampled at 300K and tested on three independent test sets for each temperature (300K, 600K, 1200K). The models can also be tested on the challenging task of computing the energy along dihedral rotations of the molecule. This test directly probes the smoothness and accuracy of the part of PES that determines which conformers are present in a simulation, and hence has a direct influence on properties of interest such as binding free energies to protein targets.

### A.2.3 Acetylacetone Dataset

The acetylacetone dataset contains trajectories of a small reactive molecule sampled at different temperature. The task is to train on snapshot sampled at 300K and test on independent test sets sampled at 300K and 600K. Moreover the extrapolation is measured both in temperature and along two internal coordinates of the molecule, the hydrogen transfer path and a partially conjugated double bond rotation, which has a very high barrier for rotation.

## A.3 Implementation Details

### A.3.1 Symmetrised One-particle Basis

For the implementation of the one-particle basis, we use `e3nn` [21] for the spherical harmonics and for symmetrising the tensor product of (8). Consequently, we also use their internal normalization.

### A.3.2 Generalized Clebsch-Gordan Coefficients

The generalised Clebsch-Gordan coefficients are defined as product of Clebsch-Gordan coefficients:

$$\mathcal{C}_{l_1m_1,..,l_nm_n}^{LM} = C_{l_1m_1,l_2m_2}^{L_2M_2} C_{L_2M_2,l_3m_3}^{L_3M_3}...C_{L_{N-1}M_{N-1},l_Nm_N}^{L_NM_N}, \tag{14}$$

where $L \equiv (L_2,..,L_N)$, $|l_1 - l_2| \le L_2 \le l_1 + l_2 \ \forall \ i \ge 3 |L_{i-1} - l_i| \le L_i \le L_{i-1} + l_i$, and $M_i \in \{m_i | -l_i \le m_i \le l_i\}$.

### A.3.3 Higher Order Features Via Loop Tensor Contractions

We implement the construction of the higher order features of Equation (10) and the message of Equation (11) in a single efficient loop tensor contraction algorithm. Below, we drop the $t$ superscript for clarity. The input variables are defined in Section 4. We give here a brief reminder, along with shape of the tensors to contract.

---

**Algorithm 1** Efficient implementation of (10) and (11) through tensor contractions of $A$-features $A_{i,klm}$ of size $[N_{\text{atoms}}, N_{\text{channels}}, (l_{\max}+1)^2]$, generalized Clebsch-Gordan coefficients $\mathcal{C}_{\eta,l_1m_1,...,l_{\tilde\nu}m_{\tilde\nu}}^{LM}$ of size $[N_{\text{coupling},\tilde\nu}] \times [(l_{\max}+1)^2]^{\tilde\nu}$, and weights $W_{z_ikL,\eta_{\tilde\nu}}$ of size $[N_{\text{elements}}, N_{\text{channels}}, N_{\text{coupling},\tilde\nu}]$ for a given correlation order $\tilde\nu$. $l_{\max}$ is the highest symmetry order of the spherical expansion of the $A$-features and $L$ is the targeted symmetry order. $N_{\text{coupling}}$ is the number of couplings of $l_1 \ldots l_{\tilde\nu}$ that yield $L$. In practice, we vectorize over $M \in \{-L, -L+1, ..., L-1, L\}$.

---

1: **function** LOOPEDTENSORCONTRACTION($A_{i,klm}$, $\{W_{z_ikL,\eta_{\tilde\nu}}\}_{\tilde\nu \le \nu}$, $\{\mathcal{C}_{\eta,l_1m_1,...,l_{\tilde\nu}m_{\tilde\nu}}^{LM}\}_{\tilde\nu \le \nu}$, $\nu$)

2: $\quad \tilde{c}_{l_1m_1,...,l_\nu m_\nu}^{z_ikLM} \leftarrow \sum_\eta \mathcal{C}_{\eta,l_1m_1,...,l_\nu m_\nu}^{LM} W_{z_ikL,\eta_\nu}$ $\quad \triangleright$ Contract coupling coefficients and weights

3: $\quad a_{i,l_1m_1,...,l_{\nu-1}m_{\nu-1}}^{z_ikLM} \leftarrow \sum_{l_\nu m_\nu} \tilde{c}_{l_1m_1,...,l_\nu m_\nu}^{z_ikLM} A_{i,kl_\nu m_\nu}$

4: $\quad$ **for** $\tilde\nu \leftarrow \nu - 1$ to $1$ **do** $\quad\quad\quad\quad\quad\quad\quad\quad\quad\quad$ $\triangleright$ Iterate over correlation orders

5: $\quad\quad \tilde{c}_{l_1m_1,...,l_{\tilde\nu}m_{\tilde\nu}}^{z_ikLM} \leftarrow \sum_\eta \mathcal{C}_{\eta,l_1m_1,...,l_{\tilde\nu}m_{\tilde\nu}}^{LM} W_{z_ikL,\eta_{\tilde\nu}}$

6: $\quad\quad \tilde{a}_{i,l_1m_1,...,l_{\tilde\nu}m_{\tilde\nu}}^{z_ikLM} \leftarrow a_{i,l_1m_1,...,l_{\tilde\nu}m_{\tilde\nu}}^{z_ikLM} + \tilde{c}_{l_1m_1,...,l_{\tilde\nu}m_{\tilde\nu}}^{z_ikLM}$

7: $\quad\quad a_{i,l_1m_1,...,l_{\tilde\nu-1}m_{\tilde\nu-1}}^{z_ikLM} \leftarrow \sum_{l_{\tilde\nu}m_{\tilde\nu}} \tilde{a}_{i,l_1m_1,...,l_{\tilde\nu}m_{\tilde\nu}}^{z_ikLM} A_{i,kl_{\tilde\nu}m_{\tilde\nu}}$

8: $\quad$ **end for**

9: $\quad$ **return** $a_i^{z_ikLM}$ $\quad\quad\quad\quad\quad\quad\quad\quad\quad\quad\quad\quad\quad\quad$ $\triangleright$ Return message $m_{i,kLM}$

10: **end function**

---

The algorithm starts at correlation $\nu$. The first step of the algorithm is to contract the generalized Clebsch-Gordan coefficients with the weights of the product basis. This contractions trades the computational cost of several products for that of a sum which is computationally very advantageous. Then, the last dimension of $\tilde{c}_\nu$ is contracted with the $A_i$-features' last dimension resulting in the $a$-tensor with correlation order $\nu - 1$. The algorithm then loops over the correlation order $\tilde\nu$ in descending order until $\tilde\nu = 1$. In each step, we first create the tensor $\tilde{c}$ by contracting the Clebsch-Gordan coefficients of correlation order $\tilde\nu$ with the weights. Then, the previous $a$-tensor is added to

the $\tilde{c}$-tensor. This operation ensures that at the end of the loop, the product basis of every correlation order are created. In fact, the $a$ contains the products for $\nu$ to $\tilde{\nu}$. The updated tensor is then contracted again with the atomic basis increasing the correlation order by 1 for all the products presented in $a$. The last $a$ tensor is exactly the message of (11).

## A.4 Tensor shapes

A glossary of the shapes of the various tensors in the MACE architecture.

| Tensor | Shapes | Equation |
|--------|--------|----------|
| $h_{i,kl_2m_2}^{(t)}$ | $[\text{N\_atoms}, \text{N\_channels}, (l_2^{\max}+1)^2]$ | (8) |
| $R_{kl_1l_2l_3}^{(t)}(r_{ji})$ | $[\text{N\_edges}, \text{N\_channels}, \text{N\_basis}]$ | (8) |
| $C_{l_1m_1,l_2m_2}^{l_3m_3}$ | $[2 \times l_3 + 1, 2 \times l_1 + 1, 2 \times l_2 + 1]$ | (8) |
| $A_{i,kl_3m_3}^{(t)}$ | $[\text{N\_atoms}, \text{N\_channels}, (l_3^{\max}+1)^2]$ | (8) |
| $\mathcal{C}_{\eta_\nu,\mathbf{lm}}^{LM}$ | $[(2 \times L + 1), [(l^{\max}+1)^2]^\nu, \text{N\_path}]$ | (10) |
| $\boldsymbol{B}_{i,\eta_\nu kLM}^{(t)}$ | $[\text{N\_atoms}, \text{N\_channels}, \text{N\_path}, (2 \times L + 1)]$ | (10) |
| $W_{z_ikL,\eta_\nu}$ | $[\text{N\_channels}, \text{N\_elements}, \text{N\_path}]$ | (10) |
| $m_{i,kLM}^{(t)}$ | $[\text{N\_atoms}, \text{N\_channels}, (L+1)^2]$ | (11) |

## A.5 Training Details

We used three codes for the paper. All MACE experiments were run with the `mace` code. All BOTNet experiments were run within the `mace` code. For NequIP experiments, we detail hereafter what code was used for what experiment. We train with `float64` precision for 3BPA and AcAc and `float32` precision for rMD17.

### A.5.1 MACE

Models were trained on an NVIDIA A100 GPU in single GPU training. Typical training time for MACE models is between 2 to 6 hours depending on the dataset. The revised MD17 models were trained with a total budget of 1,000 configurations, split into 950 for training and 50 for validation. The 3BPA models were trained on 500 structures, split into 450 for training and 50 for validation. The AcAc models were trained on 500 structures, split into 450 for training and 50 for validation. The data set was reshuffled after each epoch. We use two layers and 256 uncoupled feature channels and $l_{\max} = 3$. For all models, radial features are generated using 8 Bessel basis functions and a polynomial envelope for the cutoff with $p = 5$ [29]. The radial features are fed to an MLP of size [64, 64, 64, 1024], using SiLU nonlinearities on the outputs of the hidden layers. The readout function of the first layer is implemented as a simple linear transformation. The readout function of the second layer is a single-layer MLP with 16 hidden dimensions. We used a 5 Å cutoff for all molecules. We use the following loss function:

$$\mathcal{L} = \frac{\lambda_E}{B} \sum_b^B \left(\hat{E}_b - E_b\right)^2 + \frac{\lambda_F}{3BN} \sum_{i=1}^{B \cdot N} \sum_{\alpha=1}^{3} \left(-\frac{\partial \hat{E}}{\partial r_{i,\alpha}} - F_{i,\alpha}\right)^2, \tag{15}$$

where $B$ denotes the number of batches, $N$ the number of atoms in the batch, $E_b$ the ground-truth energy, $\hat{E}_b$ the predicted energy, $F_{i,\alpha}$ the force component of atom $i$ in the direction $\alpha \in \{\hat{x}, \hat{y}, \hat{z}\}$. $\lambda_E$ and $\lambda_F$ are weights set to $1$ and $1,000$, respectively.

Models were trained with AMSGrad variant of Adam, with default parameters of $\beta_1 = 0.9$, $\beta_2 = 0.999$, and $\epsilon = 10^{-8}$. We used a learning rate of 0.01 and a batch size of 5. The learning rate was reduced using an on-plateau scheduler based on the validation loss with a patience of 50 and a decay factor of 0.8. We use an exponential moving average with weight 0.99 to evaluate on the validation set as well as for the final model, an exponential weight decay of $5e^{-7}$ on the weights of equation 10 and 11, and a per-atom shift via the average per-atom energy over all the training set and a per-atom scale as the root mean-square of the components of the forces over the training set.

### A.5.2 NequIP

We use two implementations of NequIP for the results in the paper. We trained models on NVIDIA A100 GPU in single GPU training. Typical training time for NequIP models is between 6 hours to 2 days depending on the dataset. The results for 50 configurations rMD17 molecule were done using nequip code. The models with increasing number of layers was trained in the mace code. The timings for NequIP were also done in the mace code.

**Original `nequip` code base [5]** The NequIP model was trained on 50 configurations of rMD17 used the same model specifications as for rMD17 in [5] with the same training procedure. $\lambda_E$ and $\lambda_F$ were set to 1 and $1,000$, respectively.

**Reimplementation of NequIP in the `mace` code base** For the increasing layer experiment, the NequIP model was trained on 450 configurations and 50 configs were used for validation. We use 5 layers with 64 channels for even and odd parity, and $L = 3$ messages. We use a cutoff radius of 4Å. Radial features are generated using 8 Bessel basis functions and a polynomial envelope for the cutoff with p = 6. We use $\lambda_E$ and $\lambda_F$ weights set to 1 and $1,000$, respectively. Models were trained with AMSGrad variant of Adam, with default parameters of $\beta_1 = 0.9$, $\beta_2 = 0.999$, and $\epsilon = 10^{-8}$. We used a learning rate of 0.01 and a batch size of 5. The learning rate was reduced using an on-plateau scheduler based on the validation loss with a patience of 50 and a decay factor of 0.8. We use an exponential moving average with weight 0.99 to evaluate on the validation set as well as for the final model.

For the 3BPA timings model, we use the same model specification as in [36] with the important difference of using a polynomial envelope for the cutoff with $p = 6$ instead of $p = 2$. .