# OpenReview forum: "MACE: Higher Order Equivariant Message Passing Neural Networks for Fast and Accurate Force Fields"
_NeurIPS.cc/2022/Conference — NeurIPS 2022 Accept_

### Official Review · Reviewer_2Lfx · 2022-07-10

**Rating:** 8
**Confidence:** 4
**Soundness:** 4 excellent
**Presentation:** 4 excellent
**Contribution:** 4 excellent

**Summary:**

The paper describes a message passing NN approach that efficiently handles many-body interactions (where classical methods typically handle 2 body interactions). The method is based on a standard steerable/equivariant edge embedding and aggretation, leading to equivariant atomic environment embeddings. These per-node embeddings are then pulled through some higher order CG tensor product (like an equivariant polynomial) by which the pair-wise features start interacting with each other and effectively a higher body order is achieved. Notably, this interaction takes place on the node level (not over all possible many body combinations) which makes the method highly efficient.

The paper then systematically shows what components of equivariant architectures are important (spherical harmonic degree or max body order, or both?). To a large degree it is the max body order, and spherical harmonic degree helps but to a lesser extend. The paper further shows that with high-body order, the number of layers can be greatly reduced (to 2) whilst reaching state of the art.

**Questions:**

1. What can there be said about which observations would hold in general, in which would be limited to single (and small?) molecule analysis, versus systems of molecules (maybe like OC20?) or large molecules like proteins?
2. Dimenet and other higher-order methods like simplicial neural networks maintain a representation of the “many bodies” (e.g. dimenet evolves edge features). Another difference is the encoding of higher order interactions via geometric invariants s.a. angles, whereas MACE relies on products of invariants and covariants. I have the feeling that this product structure  (instead of e.g. summing over bodies) is important, but cannot quite put my finger on it. It is evident that the presented approach works well, but what precisely makes it successful? E.g. if Dimenet uses 3 body interactions, and this work does as well, MACE (mu=2) seems to be better, but why? Related to my puzzlement w.r.t. Dimenet comparison, if L=0 and mu=2, we get 3-body interactions through multiplication (?) of invariants like distances. Is there any interpretation to this?
3. Equation 10 reminds me of polynomical activation functions, akin to consideration in Kondor [31] that such (higher order) tensor products are non-linearities in themselves (viewing them as activation functions). I am used to the idea of such equivariant polynomial activation functions (through works like [F21]), simply as necessary non-linearities for neural networks to work. But should I change my viewpoint and consider it more than just a non-linearity, but rather see it as a way to achieve higher-body interactions?

[F21] Franzen, D., & Wand, M. (2021). General Nonlinearities in SO (2)-Equivariant CNNs. Advances in Neural Information Processing Systems, 34, 9086-9098.


**Limitations:**

Experiments are very well organized and conclusions are appropriate. It is however slightly unclear how well this method generalizes to other types of problems in computational chemistry (involving more or large molecules).

**Strengths And Weaknesses:**

**Strengths**
* The paper is very well written.
* Literature review is coherent and the storyline is clear
* Despite though material, the paper does an excellent job in presenting the method in a concise, precise and comprehendible way
* Experiments are systematically setup as to allow to specifically test individual components and in order to draw appropriate conclusions.

**Weakness**
* I found it hard to assess the impact of some of the ablation studies, in particular those pertaining to MD17.  I wonder if anything can be set about how well these results generalize to large datasets and molecules of larger size? I am wondering how representative is the aspirin molecule for the large amount of challenging problems in computational chemistry?
* The work could still benefit from a more in depth discussion on handling higher-order interactions (e.g. via simplicial neural networks or via Dimenet-like methods). It is clear that these approaches are different, is there anything to say about what difference is decisive? (e.g. eq 10 is based on products

**Small notes**
I do not see the point of the MLP in equation 13. All are linear layers, except for the last one, why? The last one is apparently a one layer MLP, is the only difference then an activation function? (otherwise a one-layer MLP == a linear layer)

---

> ### Author Response · Authors · 2022-08-02
> **Response to Reviewer 3**
>
> Thank you very much for your positive review and excellent comments. We appreciate that you find our work interesting and well-written. Below we respond to your questions and suggestions.
>
> # Use of an MLP in the last layer
>
> > I do not see the point of the MLP in equation 13.
>
> The difference between the last *readout* layer and the ones before is the *non-linear* activation instead of a *linear* one. We aim to decompose our models' output for each atom $i$ into the sum of two terms: $E_{i} = E_{i,\text{finite}} + E_{i,\text{res}}$, where $E_{i,\text{finite}}$ is interpreted as a classical polynomial body order expansion truncated up to a finite order and $E_{i,\text{res}}$ is a non-linear correction expressed as an infinite series. Since most non-linear activations have infinite Taylor expansions and, thus, do not possess a finite body-order expansion, we can only use them in the last layer. Empirically this scheme also offers the best performance.
>
> # Systems of Molecules
>
> > What can there be said about which observations would hold in general, in which would be limited to single (and small?) molecule analysis, versus systems of molecules (maybe like OC20?) or large molecules like proteins?
>
> In the past, performance on small flexible molecules at high temperatures has been a good indicator of accuracy across a large spectrum of systems \[1\]. In particular, the challenging nature of the flexible 3BPA molecule makes it an excellent proxy for performance on large proteins. We are currently applying MACE to a wide range of systems of up to 20,000 atoms, including an extensive dataset of diverse molecules, and found that MACE is performing well.
>
> # Comparison to Dimenet
>
> > Dimenet and other higher-order methods like simplicial neural networks maintain a representation of the "many bodies" (e.g. dimenet evolves edge features). \[\...\] I have the feeling that this product structure (instead of e.g. summing over bodies) is important, but cannot quite put my finger on it. Is there any interpretation to this?
>
> We want to thank you for introducing us to simplicial neural networks. As you pointed out, one central difference between MACE and DimeNet is the construction of many body representations. It is also true that MACE with $\nu=2$ and $L=0$ should be as expressive as DimeNet. However, numerous differences exist between the two models, including optimization schemes and normalization, which result in a significant difference in accuracy. We also want to stress that no higher-order models use a complete set of 4-body messages as MACE does.
>
> The way we interpret the success of MACE is twofold:
>
> -   Due to the product structure, one can construct any complete set of ($\nu + 1$)-body features at the cost of two-body ones. This alleviates the unfavorable scaling of summing over triplets and quadruplets, which prevents GemNet from creating a complete set of 4-body messages.
> -   The symmetric tensor decomposition of Eq. 10 over channels and the message passing step give an efficient re-ordering of the basis, leading to faster convergence to the complete basis limit.
>
> # Relation to Polynomical activation functions
>
> > Equation 10 reminds me of polynomical activation functions, akin to consideration in Kondor \[31\] that such (higher-order) tensor products are non-linearities in themselves (viewing them as activation functions). I am used to the idea of such equivariant polynomial activation functions (through works like \[F21\]), simply as necessary non-linearities for neural networks to work. But should I change my viewpoint and consider it more than just a non-linearity, but rather see it as a way to achieve higher-body interactions?
>
> We think both are interesting perspectives of the same operation. Seeing the tensor product as a non-linear activation in the Fourier space (like in Kondor \[31\]) would correspond to a signal processing point of view. From a physicist's point of view, they achieve an approximation of a body order expansion. The latter interpretation can help with choosing the right hyperparameters for these activations.
>
> \[1\] Dávid Péter Kovács, Cas van der Oord, Jiri Kucera, Alice E. A. Allen, Daniel J. Cole, Christoph Ortner, and Gábor Csányi. Journal of Chemical Theory and Computation, 17(12):7696--7711, 2021.

---

### Official Review · Reviewer_GriX · 2022-07-11

**Rating:** 6
**Confidence:** 3
**Soundness:** 3 good
**Presentation:** 2 fair
**Contribution:** 3 good

**Summary:**

This work proposes a new equivariant MPNN model named MACE for force field prediction. In particular, by combining equivariant message passing with efficient many-body messages, MACE achieves both state-of-the-art performance on several benchmarks and considerable computational efficiency.

**Questions:**

* For the evaluation benchmarks, what do you mean by the **revised** MD17 dataset?  Is there any difference between rMD17 and MD17 in other works? Why do the authors choose the 3BPA dataset to analyze the effect of higher order messages and the computational efficiency of the method, instead of the larger scale MD17 dataset?
* I encourage the authors to reorganize the methodology section to make it more readable and highlight their technical contributions.
* The authors should further clarify the potential negative impact of small receptive fields. If any, how do you deal with such impact?
* Can the authors provide a more formal mathematical description for the computation complexity for MACE and other baseline methods? This would be more direct to exhibit the advantage of MACE.
* Is there any typo in Eqn. (3)? The authors introduce the new symbol $h_i^{t+1}$ in line55 but don't interpret it in subsequent paragraphs.

**Limitations:**

Limitations are well discussed.

**Strengths And Weaknesses:**

Strengths:
* The proposed method achieves state-of-the-art performance on several benchmarks.
* The proposed method is faster to train and inference than previous models, which is meaningful for modeling some macromolecular systems.
* The analysis of the impact of many-body messages on learning curves is interesting, which may inspire future works in balancing message correlation orders and network depth while designing architectures.

Weaknesses:
* The connection between tensor product and the standard many-body expansion is unclear to me. The authors should provide clearer background and technical details while presenting this core module.
* The novelty of the proposed method is not well clarified. As mentioned in related works (lin99-109), both the theories and implementation strategies of the Multi-ACE framework have been put forward by previous works. There is no clear clarification of non-trivial modification in the proposed method, making it difficult to justify the technical contribution of the paper.
* The description of evaluation benchmarks is unclear. For example, the meaning of rMD17 is not well explained.
* The claim about receptive fields is misleading. In line 220 of section 5.2, the authors claim that "By decoupling the increase in correlation order of the messages from the number of message passing iterations, MACE only requires two layers resulting in a much smaller receptive field." There is no doubt that a small receptive field will make the model more parallelisable. However, the big receptive field also captures more global (long-range) interactions than the small receptive field, which usually leads to better empirical performance (see Figure 1 in this paper). It seems that the authors think the correlation order of body messages is more important than the size of the receptive field, so the negative impact of small receptive fields is ignored. But there is a lack of convincing discussion about this point. Correct me if I misunderstand something.

---

> ### Author Response · Authors · 2022-08-02
> **Response to Reviewer 2**
>
> Thank you for reviewing our paper. We appreciate your detailed questions and suggestions to which we respond below.
>
> # Connection between Product Basis and Body Ordering
>
> > The connection between tensor product and the standard many-body expansion is unclear to me.
>
> A detailed discussion of the relationship between the product basis functions and the many-body expansion can be found in Refs. \[1\] and \[2\], to which we now refer in the manuscript. The core idea is that each term of the body order expansion, which corresponds to a multivariate function, can be approximated up to any accuracy by a linear combination of product basis functions.
>
> # Relationship between Multi-ACE and MACE
>
> > There is no clear clarification of non-trivial modification in the proposed method, making it difficult to justify the technical contribution of the paper.
>
> Multi-ACE is a general framework unifying previously proposed methods. In contrast, MACE proposes a *concrete* and novel architecture and implementation with excellent accuracy and computational efficiency. To our knowledge, it is the first model to use many-body E3-equivariant messages, which we efficiently compute with a novel recursive algorithm. Finally, MACE is the first ML potential to decouple expressiveness from the number of layers. We now make this clearer in the introduction.
>
> # Description of rMD17 Dataset
>
> We have added a section in the supplementary describing the dataset in more detail.
>
> # Receptive Field
>
> > The authors should further clarify the potential negative impact of small receptive fields. If any, how do you deal with such impact?
>
> Yes, a smaller receptive field restricts the interaction range learned by the model and the effect of such restriction on the performance depends on the system. However, there is experimental evidence that current MPNNs do not capture long-range interactions beyond 8-10 Å \[4\]. However, our results strongly suggest that the need for deep MPNNs comes from the increased correlation order and not from the enlarged receptive field. For instance, Figure 1 shows that MACE with only two layers outperforms NequIP with five layers. Finally, an appealing solution to model long-range interactions could be via direct Coulomb-like interactions to complement short-range models.
>
> # MD17 vs. rMD17
>
> > For the evaluation benchmarks, what do you mean by the revised MD17 dataset?
>
> The original MD17 contained noisy data stemming from unconverged DFT calculations \[3\], and therefore, a portion of the original trajectories was recomputed with tighter convergence criteria for rMD17. Further, rMD17 contains 3 additional molecules. We now mention this in the dataset description.
>
> # Choice of the 3BPA Dataset
>
> > Why do the authors choose the 3BPA dataset to analyze the effect of higher order messages and the computational efficiency of the method, instead of the larger scale MD17 dataset?
>
> We chose the 3BPA dataset over MD17 for two reasons: (1) 3BPA is small enough to carry out extensive experiments (with 3 seeds). (2) In 3BPA, the task is to train on configurations sampled at 300K and then test on configurations sampled at 600 and 1200 K, which makes results more robust to overfitting. By contrast, MD17 consists of configurations at a single temperature.
>
> # Complexity of MACE
>
> > Can the authors provide a more formal mathematical description for the computation complexity for MACE and other baseline methods?
>
> We can estimate MACE's computational complexity by considering the costliest operations, namely the tensor product over nodes and edges. Let $N$ be the number of atoms, $N_\text{neigh}$ the average number of neighbors, $T$ the number of layers, $\nu$ the correlation order at each layer, $L_{\text{max}}$ the equivariance of the message, $N_\text{channel}$ the number of channels, and $l_{\text{max}}$ the equivariance of the edges features. Then, an estimate of the complexity is:
>
> - **MACE**: $(N_\text{neigh} ((l_\text{max}+1)^4 + (l_{\text{max}}+1)^4 (L_\text{max}+1)^2) + (l_\text{max}+1)^{2\nu} ((L_\text{max} + 1)^2 + 1)) N_\text{channel} * \mathcal{O}(N)$
>
> -  **NequIP**: $(N_\text{neigh} ((l_{\text{max}}+1)^{4} + (l_{\text{max}}+1)^{2} (L_{\text{max}}+1)^{4}) (T-1)) {N}_\text{channel}*\mathcal{O}(N)$
>
> Compare MACE with $l_{\text{max}}=3$, $L_{\text{max}}=1$, and $\nu=3$, and NequIP with $l_{\text{max}}=3$, $L_{\text{max}}=3$, and $T=5$, both with a cutoff radius of 5 Å. While the two models achieve similar accuracy, MACE is five times faster for small molecules (${N}_\text{neigh} = 17$), and 10 times faster for amorphous carbon (${N}_\text{neigh}$=80).
>
> # Clarification in Eq. 3
>
> > Is there any typo in Eqn. (3)?
>
> Thank you for pointing this out. We improved the readability of Eq. 3.
>
> # References
>
> \[1\] Dusson et al. JCP, p110946, 2022.
>
> \[2\] Christensen et al. MLST 1, 045018, 2020.
>
> \[3\] Batatia et al. 2022. https://arxiv.org/abs/2205.06643.
>
> \[4\] Nigam et al. JCP 2022, doi: 10.1063/5.0087042.

---

> > ### Comment · Reviewer_GriX · 2022-08-08
> > **Thanks for the detailed response**
> >
> > The detailed response has addressed most of my concerns. I will keep the original score.

---

### Official Review · Reviewer_FTod · 2022-07-11

**Rating:** 7
**Confidence:** 4
**Soundness:** 3 good
**Presentation:** 3 good
**Contribution:** 3 good

**Summary:**

The authors propose a new ML force field based on GNNs that uses higher-order equivariant messages. Previous methods like Dimenet++ and Gemnet use higher-order messages (triplets or quadruplets), but they enumerated the various triplets / quadruplets, which made them computationally expensive, and limited the order of interactions. In the present work, the authors use a tensor product formulation to make the framework efficient. Moreover, the authors empirically show that only 2 MP layers are needed to get good performance with their model, which makes it quite fast. SO3 equivariance is achieved using spherical harmonics representations, similar to previous methods. The authors also perform scaling laws analysis as well as extrapolation to OOD data, and show that their model performs well.

**Questions:**

* How does the cost of tensor product scale with number of nodes? In other words, can the method be applied for larger systems like materials or proteins?
* The experiments in the paper are based on small datasets. Do the authors have a sense for how MACE would perform on a much larger dataset like OC20 (opencatalyst.org)?

**Limitations:**

Seems adequate.

**Strengths And Weaknesses:**

Strengths
* The authors present a new MPNN using higher order messages. The network is equivariant which is a desirable property, for its data efficiency. The model performs well on a variety of small molecular datasets.
* The model is very fast thanks to the tensor formulation of higher order messages, and the fact that it only requires 2 layers. Efficiency for ML force fields is very useful in practice for running long MD simulations or performing structure relaxations.
* Scaling laws analysis shows desirable scaling behavior.
* The model works well for OOD data.

Weaknesses
* In section 4, the paper claims that the higher order features B_i,eta,K,L,M can be interpreted as a complete basis. Is there a reference / proof for this? The paper does not mention any.
* Experiments don't compare against newer methods like Gemnet that also use higher order message passing.

---

> ### Author Response · Authors · 2022-08-02
> **Response to Reviewer 1**
>
> Thank you for reviewing our paper. We appreciate that you find our work technically solid and of high impact. Below we respond to your suggestions to further improve the paper.
>
> Completeness of the B-Basis
> ---------------------------
>
> > In section 4, the paper claims that the higher order features $B_{i,\eta k L M}$ can be interpreted as a complete basis. Is there a reference / proof for this? The paper does not mention any.
>
> A proof of the completeness of the $B_{i,\eta k L M}$ basis can be found in Ref. \[1\]. We added the reference to the manuscript and discussed the matter in the main text.
>
> How does MACE compare to GemNet?
> --------------------------------
>
> > Experiments don't compare against newer methods like Gemnet that also use higher order message passing.
>
> We now compare our model to GemNet on the rMD17 dataset and observe that MACE outperforms GemNet (see table below for mean absolute errors on force predictions in \[eV/Ang\]). We believe that this is partially due to the inclusion of **all** four-body terms, of which GemNet includes only a subset.
>
> |  | MACE | GemNet (T/Q) |
> |---|---|---|
> | Aspirin | **6.6** | 9.5 |
> | Azobenzene | 3.0 | - |
> | Benzene | 0.3 | 0.5 |
> | Ethanol | **2.1** | 3.6 |
> | Malonaldehyde | **4.1** | 6.6 |
> | Naphthalene| 1.6 | 1.9 |
> | Paracetamol | 4.8 | - |
> | Salicylic acid | **3.1** | 5.3 |
> | Toluene | **1.5** | 2.2 |
> | Uracil | **2.1** | 3.8 |
>
>
> Scaling and Computational Cost of the Tensor Product
> ----------------------------------------------------
>
> > How does the cost of tensor product scale with number of nodes? In other words, can the method be applied for larger systems like materials or proteins?
>
> The cost of the tensor product scales linearly with the number of nodes, i.e., atoms. MACE reduces the pre-factor of the scaling compared to other methods by carrying out the most costly operations on nodes rather than edges. (For local methods, the number of edges scales linearly with the number of nodes). As a result, we could apply MACE to systems with over 20,000 atoms on a single NVIDIA A100 GPU.
>
> Applying MACE to Larger Datasets
> --------------------------------
>
> > The experiments in the paper are based on small datasets. Do the authors have a sense for how MACE would perform on a much larger dataset like OC20 (opencatalyst.org)?
>
> We are currently experimenting on a wide range of datasets containing large molecules and materials (e.g., OC20) and consistently observe good performance for MACE. In future work, we intend to discuss MACE's performance on these systems in detail.
>
> \[1\] Genevieve Dusson, Markus Bachmayr, Gabor Csanyi, Ralf Drautz, Simon Etter, Cas van der Oord, and Christoph Ortner. Atomic cluster expansion: Completeness, efficiency and stability. Journal of Computational Physics, page 110946, 2022.

---

> > ### Comment · Reviewer_FTod · 2022-08-07
> >
> > I thank the authors for their detailed response. Most of my concerns have been addressed. I will keep the original score.

---

### Author Response · Authors · 2022-08-02
**General Response**

We thank all reviewers for their time and effort in reviewing our paper. We are glad you think that the manuscript "does an excellent job in presenting the method in a concise, precise and comprehensible way" (R3) and find that our "method achieves state-of-the-art performance on several benchmarks" (R2) and "is very fast thanks to the tensor formulation of higher order messages" (R1).

In summary, we have updated our manuscript with the following changes:
-   we made the novel concepts introduced in this paper more accessible by expanding on explanations, improving the notation, and adding references,
-   we now compare our model with GemNet on the rMD17 dataset, and
-   we added a detailed description of all datasets used in this work;

Finally, we have published our code on GitHub for complete reproducibility. Below we respond to the reviewers' comments individually.

---

### Meta-Review · Area_Chair_jJbx · 2022-08-29

**Recommendation:** Accept
**Confidence:** Certain

**Metareview:**

This paper proposes a novel equivariant message passing network for modeling atomistic systems based on the popular Atomic Cluster Expansion formalism. The method relies on a clever factorization of higher order terms into products of two-body terms. This allows MACE to be fast while also taking into account many-body effects. Experimental results seem strong with intriguing scaling properties. Neural network potentials for atomistic systems is a rapidly growing subfield and this seems like a great contribution.

All reviewers supported acceptance noting the strong experimental performance, the fast training and inference speed, and demonstrated scaling with dataset size.

**Award:**

Yes

---

### Decision · Program_Chairs · 2022-09-14

Accept